# Treatment with Supercritical CO_2_ Reduces Off-Flavour of White Alfalfa Protein Concentrate

**DOI:** 10.3390/foods12040845

**Published:** 2023-02-16

**Authors:** Mikkel Hansen, Timothy John Hobley, Peter Ruhdal Jensen

**Affiliations:** National Food Institute (DTU Food), Søltofts Plads 222, DK-2800 Kongens Lyngby, Denmark

**Keywords:** green protein, white protein, sustainable foods, supercritical CO_2_, SFE

## Abstract

White alfalfa protein concentrate from alfalfa (*Medicago sativa*) is a promising substitute for milk and egg protein due to its functionality. However, it contains many unwanted flavours that limits the amount that can be added to a food without affecting its taste negatively. In this paper, we have demonstrated a simple method for the extraction of white alfalfa protein concentrate followed by a treatment with supercritical CO_2_. Two concentrates were produced at lab scale and pilot scale, with yields of 0.012 g (lab scale) and 0.08 g (pilot scale), of protein per g of total protein introduced into the process. The solubility of the protein produced at lab scale and pilot scale was approximately 30% and 15%, respectively. By treating the protein concentrate at 220 bar and 45 °C for 75 min with supercritical CO_2_, off-flavours were lowered. The treatment did not decrease the digestibility or alter the functionality of white alfalfa protein concentrate when it was used to substitute egg in chocolate muffins and egg white in meringues.

## 1. Introduction

As the global population increases, so too does the demand for food. Currently, many new plant-based protein sources are being investigated as a means to meet the growing demand for food, while maintaining or lowering the total emission of greenhouse gases.

Since 2009, alfalfa protein concentrate (APC) has been regarded as safe for consumption as a food supplement in the European Union at concentrations up to 10 g/day [1]. However, the APC is not of high enough quality to substitute other alternative proteins, such as soy protein, which is the main plant-based protein on the market [2]. One of the problems with APC is the high fraction of fibre (Table 1), which consists mainly of hemicellulose and cellulose and a smaller fraction of lignin [3]. The high indigestible fibre fraction, which is known to bind proteins, is likely the reason why the digestibility of APC is lower than soy protein isolate (SPI), soy meal flour (SMF) and the white protein concentrate “Welpro” [4].

Alfalfa is a perennial plant, grown worldwide as a feed for ruminants, pigs and hens. In the last decades, alfalfa has been of increasing interest as a new source of protein for human consumption. This is due to its amino acid composition, which contains all essential amino acids and the high protein yield per acre of land [11,12,13]. Alfalfa is typically harvested 4–5 times per year and made into silage or transported directly to a biorefinery for protein extraction for animal feed. Extraction is typically carried out through a mechanical separation of green juice (protein rich wet fraction) from the pulp (fibre rich solid fraction). The green juice is subjected to further processing, during which the proteins can be precipitated and recovered by centrifugation. The green protein pellet is collected as the product. The supernatant, now called the brown juice, is typically used for biogas or fertilizer [3,4,11,14,15,16] even though it is rich in protein.

The green alfalfa protein concentrate looks, smells, tastes of grass and it is usually used as animal feed [17]. Nevertheless, alfalfa protein concentrates have been proposed to be incorporated into foods for several years. In all cases, the bitterness or “grass” taste has been a challenge for the consumer, and this has limited the amount of concentrate that could be included in the food formulation [7,18,19]. Studies have shown that the main contributor to this bitter taste is the saponin, zahnic acid, which is one the largest groups of saponins in alfalfa [20,21,22]. Other studies have shown that extraction of saponins from alfalfa is possible with supercritical CO_2_ extraction (SFE), a method which is known to consume low amounts of energy and have the ability to preserve the functionality of the material extracted [23,24].

The main protein in alfalfa is RuBisCo, which constitutes 30–70% of the total protein content [25]. RuBisCo is one of the largest proteins found in nature (up 560 KDa), it contains all essential amino acids and it is known for its functionality with respect to emulsifying and foaming capabilities [11,26]. Due to its amino acid profile and functionality, RuBisCo is of high interest as a vegetable substitute for animal derived agents. RuBisCo is highly reactive and it is easily oxidized, which lowers its functionality. Thus, when working with this protein, limiting its oxidation is of high importance [27].

In 1975, a complex process, called the Pro-Xan II method, was proposed to produce edible white alfalfa protein concentrate (WAPC). The process consists of more than 19 processing steps and recovers roughly 0.08 g of purified white protein (Welpro in Table 1) per g of protein input in the process [7]. In a recent study, it was proposed to use ultrafiltration/diafiltration of the brown juice, instead of using acid precipitation, to obtain a WAPC with a protein concentration of 60.5% (DM). This was followed by an adsorptive removal of polyphenols, thus requiring a rehydration of the WAPC [28]. In 2015, Faris-Campomanes et al. demonstrated that SFE could be used for the extraction of polyphenols from lees [29]. Therefore, it seems possible that SFE could be used to extract saponins, and polyphenols from alfalfa protein concentrates.

In this study, we propose and investigate a new simple processing method to produce WAPC from brown juice. The entire process from raw alfalfa to WAPC consists of only eight steps including SFE, to produce a WAPC with acceptable taste, composition and functionality to substitute animal proteins in baked goods.

## 2. Materials and Methods

Two different batches of WAPC were produced during this study, one batch where a laboratory size twin screw press was used to press the alfalfa and one where a larger process scale single screw press was used. In order to differentiate between the different scales of production and whether the WAPC was treated with supercritical CO_2_ or not, the nomenclature shown in Table 2 is used.

### 2.1. Production of White Protein Concentrate

Alfalfa was harvested in early spring, on Fyn (Denmark), manually with a scythe, packed in bags, then transported directly to the laboratory (within 3 h) and stored at −20 °C until it was further processed with a twin or single screw press. Before processing, roughly 200 g of raw alfalfa was withdrawn for composition analyses (protein, moisture, ash and fat) as described in Section 2.2. To recover white protein concentrate, the alfalfa was first thawed by submerging it in cold tap water, followed by draining it. The plant was then processed through a twin screw press (Angel Juicer S8500 (Seoul, Republic of Korea)) or a single screw press (Vincent CP-1 (Vincent Corporation, Tampa, FL, USA)) resulting in two fractions; a protein rich green juice and a pulp fraction. For large-scale trials, the pulp was re-suspended with 2× its own weight of water before being pressed through the single screw-press again, resulting in a second fraction of green juice that was then pooled together with the first fraction. The green juice was continuously collected and stored at 4 °C to minimize potential oxidization of the proteins.

In order to determine the best temperature for green protein precipitation from the green juice, a simple test was first carried out. The pH was adjusted to 8 and the green juice was heated to 45, 50 or 55 °C at a rate of >2 °C/min. When it had reached the test temperature, it was centrifuged at 2500× *g* for 20 min at 4 °C (Thermo Scientific Multifuge X3R, Waltham, MA, USA). The green protein precipitate was removed and the soluble protein remaining in the supernatant (i.e., the protein depleted brown juice) was measured. The lowest amount of soluble protein left in the protein-depleted brown juice occurred at 50 °C. Based on those findings, it was decided to continue with 50 °C for removing the undesirable green protein fraction while maintaining the highest amount of soluble protein in the white protein fraction (see Appendix A for data).

The pH of the collected brown juice was lowered to 3.5 by addition of 3M lactic acid, while mixing it and then it was immediately centrifuged (2500× *g*, 20 min, 4 °C). The white protein pellet was then collected and freeze-dried (Thermo Scientific Heto Drywinner DW8, Waltham, MA, USA), starting at −20 °C with a 1 °C/h increase until the temperature reached 20 °C where it was kept, until no decrease in weight was observed. To ensure a uniform distribution during SFE treatment, manual crushing to a particle size below 1.25 mm (particle size distribution in Appendix C) was carried out with mortar and pestle. This resulted in a white alfalfa protein concentrate (WAPC). A simplified illustration of the process can be seen in Figure 1.

### 2.2. SFE Treatment of WAPC

A preliminary experiment was first conducted to determine the operating conditions for the SFE system. The WAPC was dried at room temperature in a vacuum oven over night (VT 6060, Thermo Fisher Scientific, Waltham, MA, USA). Then samples of 10 g were placed in 25 mL extraction vessels. Extraction was carried out for 75 min using either 180, 220 or 260 bar, with or without 1 or 2 mL/min of co-solvent, using the following conditions: 45 °C, 10 mL/min CO_2_ flow, 60 min dynamic time, 15 min static time, make up flow 0.5 mL/min of 96% ethanol. The temperature of 45 °C was used to minimize solubility loss of the protein based on the test illustrated in Appendix A. A MV-10 (Waters, Milford, CT, USA) SFE system was used. *Chlorophyll* in the extract was estimated by spectrophotometry (Genesys 10S, Thermo Fisher Scientific, Waltham, MA, USA), using the equation described by Mouahid et al. [30]:$$\begin{array}{c} Chlorophyll_{a}=12.74\cdot A_{663}-2.69\cdot A_{645}\\ Chlorophyll_{b}=22.9\cdot A_{645}-4.68\cdot A_{663}\\ Chlorophyll_{total}\left(µg/mL\right)=Chlorophyll_{a}+Chlorophyll_{b} \end{array}$$

The level of grass aroma was observed by sniffing the protein powder after extraction. The results showed (See Appendix B) that a combination of 220 bar without the use of co-solvents at 45 °C led to the best compromise between removal of *chlorophyll* and grass aroma. These conditions were used in all future work.

To extract enough WAPC for baking experiments, four samples of 10 g were placed in four 25 mL extraction vessels, and they were extracted using the optimal conditions described above. The four extracts were pooled and then the absorbance spectrum from 300–600 nm was measured (Genesys 10S, Thermofischer Scientific, Waltham, MA, USA) against a blank with 96% ethanol to inspect for the removal of colour compounds during extraction.

### 2.3. Moisture and Ash Content

Pre-dried crucibles were filled with roughly 3 g of sample and left in an oven at 104 °C overnight. The crucibles were then cooled in a desiccator before being weighed again. Moisture content was found from the loss of weight after drying. The raw alfalfa was treated in a similar way; however, due to the sample size of roughly 200 g, only one analysis was conducted for this. The dried alfalfa was then blended to a uniform powder, re-dried, and hereafter treated as the other samples for ash measurement.

After the moisture content was determined, the samples were placed in a muffle furnace at 600 °C overnight (D6450 M110, Heraeus, Hanau, Germany) and then placed in a desiccator for cooling. The weight of the cooled incinerated crucibles was then used to determinate the ash content.

### 2.4. L-a-b Colour Measurement

Around 1 g of sample was placed under a glass plate and L-a-b colour was measured using a LC 100 spectrocoloriometer (Lovibond, Amesbury, UK). Three measurements were carried out for each sample.

### 2.5. Total Fat Analysis

Total fat content was found by using the Rapid NMR Fat Analyzer (CEM, Matthews, NC, USA) with the Powder method. All analyses were carried out in biological duplicates.

### 2.6. Effect of pH Change on Protein Solubility

To investigate optimum solubility of the WT (Table 2), it was analyzed at different pH values. For this, 50 mg of WT was mixed with 2 mL of 0.1 M sodium phosphate buffer at different pH values of 7, 8, 9, 10 or 11 by vortexing for 20 s. They were then mixed in a laboratory shaker for 10 min at 1000 rpm (TS-100C, Biosan, Riga, Latvia), to ensure uniform distribution of the WAPC. After shaking, the samples were centrifuged for 5 min at 10,000× *g* and room temperature in a Microcentrifuge (Ole Dich, Hvidovre, Denmark). After centrifugation, the pH of the supernatant was measured using a LAB 845 pH meter (Xylem, Mainz, Germany). To measure the effect of pH changes on solubility of the WAPC, the concentration of soluble protein was determined using the Bradford analysis method (described in Section 2.7).

The effect on protein solubility of lowering the pH after it had been raised was also studied. In this case, the pH of WAPC was raised to values of 9, 10 and 11 as described above. The supernatant was recovered after centrifugation and then it was solubilized in a sodium phosphate buffer at pH 8, followed by centrifugation and analysis of the supernatant for soluble protein.

### 2.7. SDS-PAGE

All reagents and equipment used in these analyses were from Biorad (Hercules, NJ, USA). Samples were prepared by mixing 50 mg of WT with 2 mL of 0.1 M sodium phosphate buffer pH 8 (10 min, 1000 rpm (TS-100C, Biosan, Riga, Latvia)), then centrifuged for collection of the supernatant (5 min 10,000 rpm (Ole Dich, Hvidovre, Denmark)). An amount of 10 µL of the supernatant was then mixed together with 5 µL 4× Laemmli buffer, 4.75 µL of milliQ water and 0.25 µL of β-mercaptoethanol. The solution was then incubated for 10 min at 95 °C in a TS-100C heating block (Biosan, Latvia, Riga). An amount of 10 µL of the incubated sample was then loaded into a well in a 4–20% Mini-PROTEAN^®^ TGX gel where the first well was loaded with 5 µL Precision Plus Standard ladder. The gel was run at 140 V, 400 mA for 50 min. The gel was then washed and stained for 1 h (Coomassie R-250) followed by a de-staining procedure. This consisted of replacing the staining reagent with water and leaving the gel with gentle shaking for 1 h, before replacing the water with fresh water. The last procedure was repeated 3 times before scanning the gel using a ChemiDoc XRS + System.

### 2.8. Protein Determination

Soluble protein was measured using the Pierce Coomassie Plus Bradford kit (Thermofischer Scientific, Waltham, MA, USA) following their protocol for using microwell plates [31]. All analysis was carried out in triplicate on an Infinite M200 Pro plate reader (TECAN, Männedorf, Switzerland).

Total protein was determined by using the DUMAS combustion method with a rapid MAX N exceed (Elementar, Langenselbold, Germany). Protein was calculated by using a protein factor of 6.25 [7].

### 2.9. Protein Digestibility

Protein digestibility was found by following the pepsin–trypsin method described by Saunders et al. [10]. First, 1 g of sample was suspended in 20 mL 0.1 M HCl containing 50 mg of trypsin (dissolved in 1 mL of 0.01 M HCl) and incubated for 48 h at 37 °C with gentle shaking. This was followed by centrifugation (2500× *g*, 15 min, 4 °C) using a Thermo Scientific Multifuge X3R (Waltham, MA, USA). The pellet was then suspended in 10 mL of distilled water and 10 mL of 0.1 M sodium phosphate buffer (pH 8.0) containing 5 mg of trypsin was then added. This was incubated at 23 °C for 16 h before being centrifuged (2500× *g*, 15 min, 4 °C). The solids were then washed 3 times by suspending with 30 mL of distilled water each time and centrifuging (2500× *g*, 15 min, 4 °C), except for the last wash in which the solids were filtered through a 1.2 µm pore sized nitrogen-free filter. The filter was then dried in a VT 6060 vacuum oven (Thermofischer Scientific, Waltham, MA, USA) at room temperature, overnight. The digested dried pellet, including the filter, was then analyzed for total protein content as described above and the protein digestibility was found by comparing total protein content before and after digestion.

### 2.10. Foam Stability of WS and WSS Compared to Milk

To mimic the low fat content in WS and WSS, a semi-skimmed milk (Egelykke, Arla-foods, Viby, Denmark) with 3.5% protein and 1.5% fat (data from the ingredient list) was purchased in a retail shop. A 23.75 mL aliquot of the milk was transferred to a 50 mL plastic centrifuge tube. Samples of WS and WSS were dissolved in sodium phosphate buffer (pH 8.00) to a protein concentration of 3.5% and a final volume of 23.75 mL, in a 50 mL plastic centrifuge tube. The different solutions were then mixed vigorously for 10 s at 10,000 rpm with a T-25 Ultra Turrax disperser (IKA, Staufen, Germany) to create foam. Foam height and foam strength were measured with a TA.XT.plus texture analyzer (Staple Micro Systems, Godalming, UK), using a 25 mm cylinder probe. The test speed was 1 mm/s at a distance corresponding to the top of the liquid. The average strength of the foam was calculated based on the average force (N) required to lower the probe from the foam top until the liquid layer. Foam height was measured from the top of the foam to the liquid layer.

### 2.11. Production of Meringues

Three batches of meringues were produced to investigate the foaming capabilities of the WAPC when it was included in a food. The basic recipe was 1.6 g of powdered sugar per g of egg white, where the egg white was replaced by WT or WTS as appropriate. Pasteurized egg white (Dava Foods, Hadsund, Denmark) was purchased in a retail store, which contained 9.1% protein and 90% moisture (wet weight). WT and WTS were used to make an egg replacement by mixing an appropriate mass, to give the same protein content as egg white, with sufficient deionized water (pH 8.00 adjusted with NaHCO_3_) to match that in egg white.

The pure egg white, WT or WTS egg replacements were foamed at 15,000 rpm with a T-25 Ultra Turrax disperser (IKA, Staufen, Germany), while powdered sugar was slowly added. After foaming, the three different batches were distributed in silicone moulds with cylindrical wells (35 mm diameter, 0.9 mm height), to a level just below the top of the form. They were then baked for 45 min at 85 °C, with low ventilation, in a Rational C11C95057013 Combi Oven (Rational, Landsberg am Lech, Germany) followed by 15 min at 120 °C. The meringues were then cooled for 20 min before further analyses.

### 2.12. Production of Chocolate Muffins

Four different batches of chocolate muffins were produced to demonstrate the WAPC functional properties: one batch without egg or WAPC as a control, one with egg, one with WS and one with WSS. In the two batches with WAPC, the amount of WAPC corresponded to the amount of protein added from eggs in the recipe and additional water was added to make up for the moisture content in the egg. The water content of egg was calculated based on data from the Danish National food database [32]. The muffin dough consisted of 60 g wheat flour, 50 g sugar, 1.5 mL baking soda, 1.5 mL vanilla sugar, 30 mL cocoa, 50 g egg and 30 g water. The ingredients were mixed together to give a uniform dough. Subsequently, 15 mL of the dough was placed in a paper muffin mould (diameter 50 mm, height 35 mm) and baked at 180 °C for 20 min with low ventilation in a XVC 705 oven (UNOX, Cadoneghe, Italy) and left to cool for 20 min before further analyses.

### 2.13. Texture Analysis of the Chocolate Muffins and Meringues

The texture of the muffins and meringues was analyzed with a TA.XT.plus texture analyser (Staple Micro Systems, Godalming, UK). For both, springiness and chewiness were measured with a flat disc (100 mm diameter) with the following settings: test and post speed of 5 mm/s, strain at 75%, post waiting time of 5 s and trigger force of 0.049 N.

For the meringues, height and hardness were measured with the same disc and settings as for springiness and chewiness.

For the muffins, height and hardness were analyzed with a knife cutter in the same apparatus. Travel speed was set to 1 mm/sec and travel distance was set to 20 mm from the top of the muffin. Height was measured as the distance from the top of the muffin to the bottom plate.

### 2.14. Sensory Analysis of the Meringues

A focus group was used, consisting of 19 different persons. The participants consisted of 12 females and 7 males ranging from 18 to 62 years in age. They were asked to rank the two batches of meringues with respect to which one had the strongest grass taste. Before tasting the test meringues, the participants took a bite of the control meringue with egg white and drank a sip of water. The participants were all made aware that the meringues contained WAPC before the test, but not which batch contained the SFE-treated WAPC.

### 2.15. Sensory Analysis of the Chocolate Muffins

A focus group was used, which consisted of 11 employees from DTU Food. The participants ranged from 20 to 62 years of age and there were five males and six females. The focus group was asked to taste the four types of chocolate muffin and to rank them on a taste of grass scale (no grass taste, mild grass taste, strong taste). After the participants had ranked the muffin, a discussion about the muffins was carried out in plenum.

### 2.16. Statiscal Analyses

Each analytical result reported is the mean value of three replicate measurements or biological triplicates as stated in the captions to the data figures or tables, unless otherwise reported. Standard deviation and statistical differences were analyzed in MS Excel. Differences between the means were analyzed using the single factor ANOVA test with a least significant difference of 0.05.

## 3. Results

The overall aim of this work is to examine if white alfalfa protein concentrate is a suitable substitute for eggs in baked goods. This requires that the protein produced has a suitable amino acid profile, digestibility and functionality and that undesirable sensory properties can be satisfactorily ameliorated, in this case using supercritical CO_2_ extraction.

### 3.1. Yield and Protein Concentration of the WAPC Obtained from the Single and Twin Screw Presses

From 4 kg of wet alfalfa, 17.5 g of WT was obtained, with a protein concentration of 57% using a twin screw press as shown in Table 3. This resulted in a yield of 0.012 g of protein per g of protein introduced into the twin screw press. The total protein content was found to be higher in the WTS, whereas no fat was present in the WTS. The colour in both WT and WTS was similar.

In order to generate enough WAPC for trials with muffins, a pilot scale production was made. Yield of the WAPC production was not part of the scope of this study, nevertheless, when 22.2 kg (wet) of alfalfa was used, 102.2 g of dry WS was recovered (6.17% moisture), which had a protein concentration of 57.73%. The protein concentration in the raw alfalfa was found to be 4.31% on a wet basis (derived from Table 4). This means that the yield was 0.06 g of protein per g of protein introduced into the screw press. However, this yield results from pooling the two fractions obtained from the first pressing of raw alfalfa, and the second pressing of the pulp. Since the main part of the plant material is lignocellulosic fibre, the rest of the WAPC is speculated to be mainly fibre and free carbohydrates; however, this was not investigated in the current study. The yield from the single screw press is 0.02 g/g lower than the method proposed by Edwards et al. in 1975 comprising 19 processing steps [7]. However, upon cleaning the screw press, it was noted that approximately 1.5 kg of pulp was left in the filter matrix due to its design.

Supercritical fluid extraction can be expected to remove small molecular weight polar compounds and fats from the WAPC in addition to unwanted flavour and aromas [33]. After SFE treatment, the protein content of the WSS was found to have increased slightly to 63.6%, which is expected to be due to the removal of fats (Table 4) and other non-protein compounds. A similar effect was seen for the twin screw press (Table 3). It has previously been shown that SFE can be used for the extraction of *chlorophyll* (green colour) and xanthophyll (yellow colour) from other plant materials [34,35]. As therefore expected, the SFE treatment reduced the green and yellow colour in the WSS, as shown by the LAB measurements in Table 4 and increased the colour in the extract recovered after the SFE treatment. The SFE extract had a visible yellow-green colour (OD_580(Yellow)_ = 1.41, OD_550(green)_ = 1.08), which is likely to be *chlorophyll* and xanthophyll pigments [15,33,36]. It should be noted that these measurements were made on the extract, which had been mixed with 35 mL of ethanol (96%) from the make-up flow during the extraction. The colour of the WAPC produced with the twin screw press was lighter, greener and less yellow (Table 3) compared to the WAPC produced with the single screw press.

No difference was observed with respect to *chlorophyll* extraction when the pressure was increased from 220 to 260 bar. Therefore, 220 bar was chosen to minimize the potential cost for a full-scale production. Future studies should include WAPC extracted at higher pressures to investigate if this could lower the taste of grass for the consumer, while maintaining the functionality of the WAPC.

The digestibility of the protein was not affected by the SFE treatment as seen in Table 4. WSS had higher average protein concentration, better digestibility and lower fat compared to the APC in Table 1.

### 3.2. Solubility of the Proteins

The WT is at pH 3.5 after precipitation, and it was therefore of interest to see if solubilization could be enhanced by raising the pH. This was therefore tested at various pH values. From Figure 2, it can be seen that the protein was most soluble when raised to pH 8. Interestingly, when the pH was lowered to 7 from pH 8 or 8.5, the concentration of soluble protein was higher than if the protein had been raised to pH 7 in one step (Figure 2). The protein was therefore brought to a pH of 8 for the subsequent studies.

The pattern of solubility seen in Figure 2 was also reflected in the results from analysis by SDS-PAGE (Figure 3), where intense bands were observed in lane 4 (i.e., pH 8), which also had the highest solubility in Figure 2. The least intense bands were observed for lanes 1 and 5, which had low solubility as well. Interestingly, lane 8 had the strongest bands in the gel, which suggest that more of the RuBisCo fractions were solubilized by increasing pH to 8.5 before lowering it to a neutral pH. Both the large and the small subunit of the RuBisCO protein were observed in all the lanes (55 kD and 14 kDa, respectively) [27]. The bands around 40 kDa are most likely the degraded species of the large subunit of RuBisCO [37].

The protein solubility of the WSS (15.80%) was observed to be 0.72 percentage points higher, which was not significant, compared to the untreated WS (14.98%). This is confirmed by the SDS-PAGE analysis in Figure 4, where there were only small differences observed between the two protein concentrates. The overall solubility of the WS was half of the WT (compare Table 4 and Figure 2). This is also illustrated in Figure 4 where the two RuBisCo fractions (around 55 Kd and 14 Kd) was less intense compared to the bands observed on the gel prepared with WAPC from the twin screw press (Figure 3). It could be speculated that the processing conditions in the single screw press increased oxidation of the proteins and thereby lowered the solubility of those proteins.

### 3.3. Foam Stability of the Proteins

The foaming properties of proteins are important functional attributes in food production. When a solution of WS or WSS was compared to semi-skimmed milk, it was seen that the WS and WSS (Table 5) produced a greater height and strength of foam. Furthermore, the SFE treatment had no negative effect on foaming properties. It was observed that 2 h after the test, more than 20 mm of foam was still present for both WS and WSS. In comparison, the semi-skimmed milk foam had dissipated within 2 min. The results here are consistent with previous work showing that soluble leaf protein from alfalfa has useful foaming capabilities [37,38,39]. Based on these results, it was decided to continue with developing batches of meringues to further study WAPC as a foaming agent.

### 3.4. Texture Analysis of the Meringues

Three batches of meringues were produced as described in Materials and Methods. Even though the mass of the respective doughs was the same, the volume of the doughs made with WAPC was slightly lower compared to the control with egg white. This is illustrated by the number of produced meringues in the moulds in Figure 5A–C. This suggests a higher density of the doughs made with WAPC compared to the one with egg; however, this was not investigated further.

During cooking, the meringues with WAPC raised uniformly as can been seen in Figure 5E,F (the cracked WAPC meringues were broken manually after removal from the oven), whereas the standard meringues with egg white raised unevenly and cracked at the end of cooking (Figure 5D). The visual appearance of the dough made with WT and WTS (Figure 5B,C) was similar but was darker than the white appearance of the meringue made with egg white (Figure 5A).

The height of the meringues was measured from the point where the test probe registered resistance. The control with egg white had a cracked surface with a conical-like shape; therefore, the average height of the egg white meringues was in reality lower than the measured height presented in Table 6. The height of both meringue types produced with WAPC was almost identical, but the hardness of the crust was significantly higher for WTS compared to the WT (*p* > 0.05). The hardness of the WT meringue was lower than the other two batches, and this is speculated to be due the higher fat content, as described in Table 3. However, further investigation is needed to confirm this. The hardness of WTS meringues was not significantly different from that of the standard with egg white.

### 3.5. Sensory Analysis of Merringues

All of the participants in the focus group could taste grass in the meringues produced with WAPC. However, 18 out of 19 participants found that the WTS meringue had the mildest taste of grass compared to the WT (presented in Table 7). The one participant who thought that the WT had the mildest taste explained that he thought that the taste of grass was milder in the WTS but lasted for a longer time, whereas the WT meringue had a strong taste of grass in the beginning of the taste experience and then quickly faded. Some of the participants liked the taste of the WTS meringue and others suggested masking the mild grass taste with ingredients such as ginger and liquorish.

### 3.6. Application of WAPC in Chocolate Muffins

The WS and WSS was examined to see if it could replace egg in baked goods. In light of the results from the meringue sensory testing, chocolate muffins were chosen as a test system, since the chocolate colour and flavour would be expected to mask any grassy notes or colour changes. In general, the results in Table 8 show that muffins with WS and WSS did not have a texture that was comparable to muffins with egg. The WS- and WSS-containing muffins were, in fact, not significantly different to the muffins without egg (Table 8). Nevertheless, muffins with egg and WS and WSS had all risen more than the control without egg (Table 8), although an ANOVA test shows that this is not significant (*p* = 0.11). From these data, it is clear that the standard muffin with egg had the best parameters with respect to its texture. As can be seen in Figure 6, the visual appearance of the four muffin types was quite similar. Saponins are known for their gel-forming capabilities, and it has been proven that most of the saponins are recovered in the green pellet and not in the WS and WSS [40,41]. This could be an explanation of the lower springiness observed in the two batches of muffin with WS and WSS compared to those with egg. However, the saponin content was not measured and future studies should be carried out to verify this.

### 3.7. Sensory Analysis of WAPC in Muffins

In order to determine whether the chocolate flavour was a way of masking the grass taste, the muffins were tested by a focus group. The main finding from the focus group analysis of the muffins was that the SFE treatment of the WSS lowered the taste of grass, compared to the muffins containing WS, which had not undergone SFE treatment as seen in Table 9.

All participants in the focus group noted a mild grass taste in the muffin with WSS and a strong taste of grass in the muffin with WS. In the discussion in plenum, the muffin with WSS was mentioned as acceptable to chew, but they would have preferred if the taste of grass could be lowered. The muffin with WS was unpleasant overall and some found it disgusting. They mentioned that the control without egg was unpleasant to bite in and that the standard muffin with egg was pleasant overall. None of the participants in the test tasted any grass in the control muffin without egg and the standard with egg. No one observed any colour differences in the four muffin types.

## 4. Discussion

In this work, we have focused on two aspects for producing a protein ingredient from alfalfa: the design of a simple method of extraction for white protein concentrate and an off-flavour removal step that does not affect the functionality of the protein. Compared to the method proposed by Edwards et al. 1975 [7], our process is much simpler (9 compared to 19 steps), but gives a slightly lower yield of WAPC when using the large single screw press (0.06 g/g versus 0.08 g/g). Most of this is thought to be due to a loss of protein in the screw press due to the large dead volumes compared to the amount of alfalfa processed, since a yield of 0.12 g/g was possible with the smaller twin screw press. Some of the protein is recovered as green protein, which can be used for feed, but a large fraction is left unrecovered in the pulp. It is likely that much of the un-extracted protein is non-RuBisCo protein and thereby of a lower quality than the protein extracted. Future studies should investigate the composition of the remaining protein in the pulp to clarify whether or not developing a method for a higher yield of WAPC would be economically relevant.

The protein concentration in our process (57%) was around 30% lower compared to Edwards et al. (89%); however, the protein digestibility was similar (93% versus 92%). This was expected due to our simplified method. Even though it would be possible to refine and increase the protein concentration of the WAPC further by, for example, ultrafiltration and or acid washing, the potential cost of these procedures could make the resulting WAPC too expensive to compete with other vegetable protein concentrates on the market.

RuBisCo is known for its ability to emulsify and foam. However, the performance of WAPC as a substitute for egg, as an emulsifying agent was not found to be optimal. The low saponin content in WAPC is thought to be the reason for this. However, using WAPC as a substitute for foam-creating agents, such as egg white, was found to be favourable.

The WAPC we produced with a twin screw press had a higher solubility and our SDS-PAGE analysis had more intense bands with respect to the RuBisCo fractions in the WAPC produced with the single screw press. This suggests that a potential production of WAPC in full scale should be conducted with a twin screw press or in a setting where potential oxidation of the green juice during processing is minimized to maintain the highest levels of functionality in the RuBisCo protein.

## 5. Conclusions

We have demonstrated that our simplified nine-step extraction method, including a supercritical CO_2_ step, can be used to produce white alfalfa protein concentrate with levels of grass taste and colour that are acceptable for substituting egg white as a foaming agent in baking.

## Figures and Tables

**Figure 1 foods-12-00845-f001:**
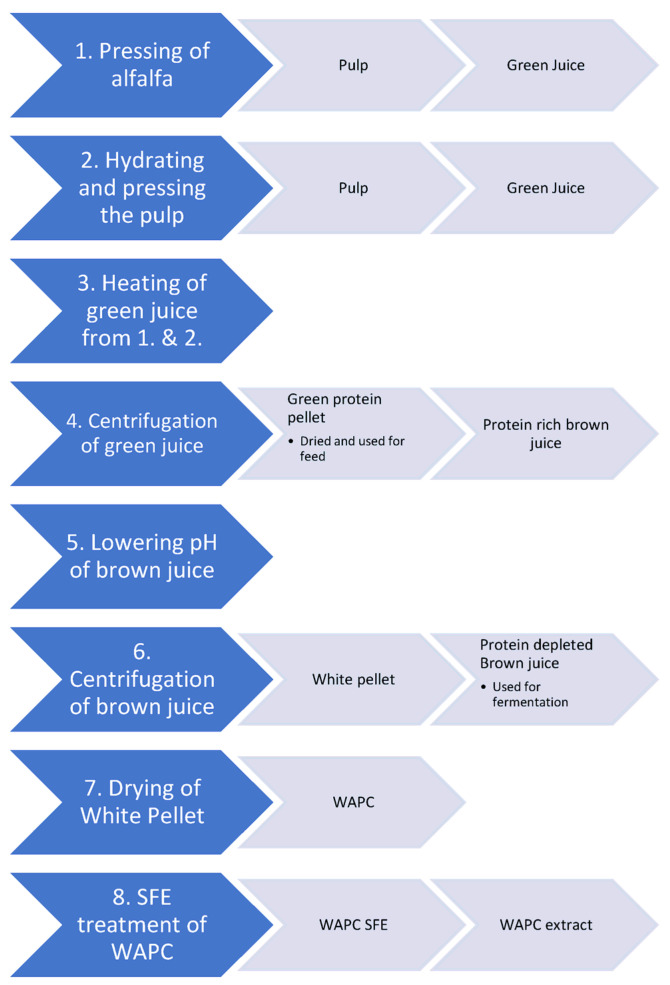
Process used for production of white alfalfa protein concentrate. Blue arrows = processing step, grey arrows = products from the processing step. WAPC = white protein concentrate, SFE = supercritical CO_2_ extraction. When using the small twin screw press, step 2 was omitted.

**Figure 2 foods-12-00845-f002:**
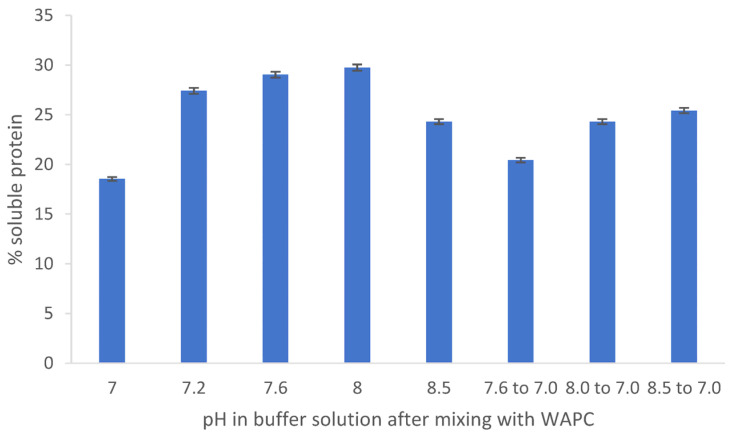
Percentage of soluble protein in the WT when suspended in sodium phosphate buffer solutions at various pH values. In the three bars to the right, the pH was decreased back to 7.0 after it had first been raised to a higher pH, *n* = 3.

**Figure 3 foods-12-00845-f003:**
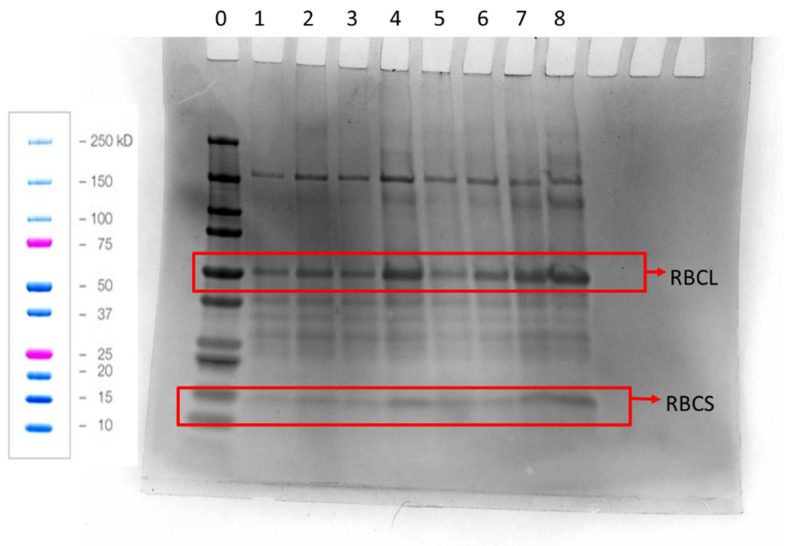
Reduced SDS-PAGE gel of WT, from the twin screw, solubilized in sodium phosphate buffer solution at various pH values (lanes 1–5) or raised to high pH and then lowered to neutral pH (lanes 6–8). Lane 0 = molecular weight standards, lane 1 = pH 7, lane 2 = pH 7.2, lane 3 = pH 7.6, lane 4 = pH 8, lane 5 = pH 8.5, lane 6 = pH 7.6 to 7, lane 7 = pH 8.0 to 7, lane 8 = pH 8.5 to 7. RBCL = large RuBisCO subunit, RBCS = small RuBisCO subunit. The figure on the left with blue and pink coloured bands show Biorad’s instructions of how the standards are expected to run. The same mass of WAPC was used to prepare the sample loaded into each well.

**Figure 4 foods-12-00845-f004:**
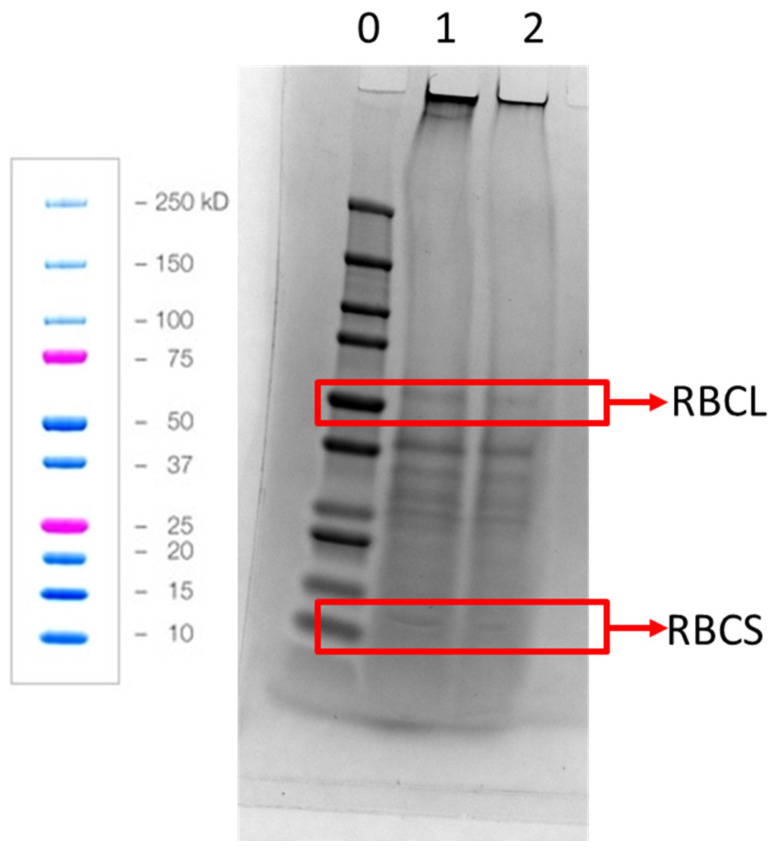
Reducing SDS-PAGE of the WS and WSS. Lane 0 = molecular weight standards, lane 1 = WS, lane 2 = WSS. RBCL = large RuBisCO subunit, RBCS = small RuBisCO subunit. Diagram on the left with blue and pink coloured bands is from Biorad’s instructions showing how the standards are expected to run. The same mass of WAPC was used to prepare the sample loaded into the wells.

**Figure 5 foods-12-00845-f005:**
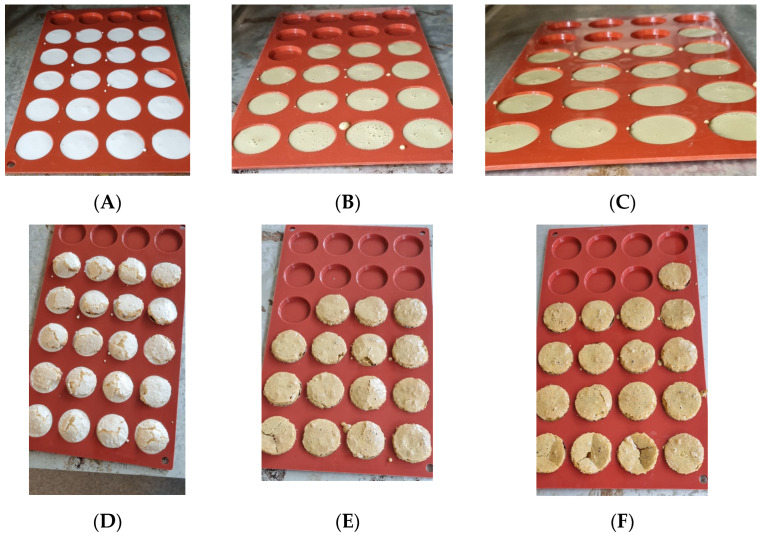
Pictures of the meringues before (**A**–**C**) and after cooking (**D**–**F**). Egg white (**A**,**D**), WT (**B**,**E**), WTS (**C**,**F**).

**Figure 6 foods-12-00845-f006:**
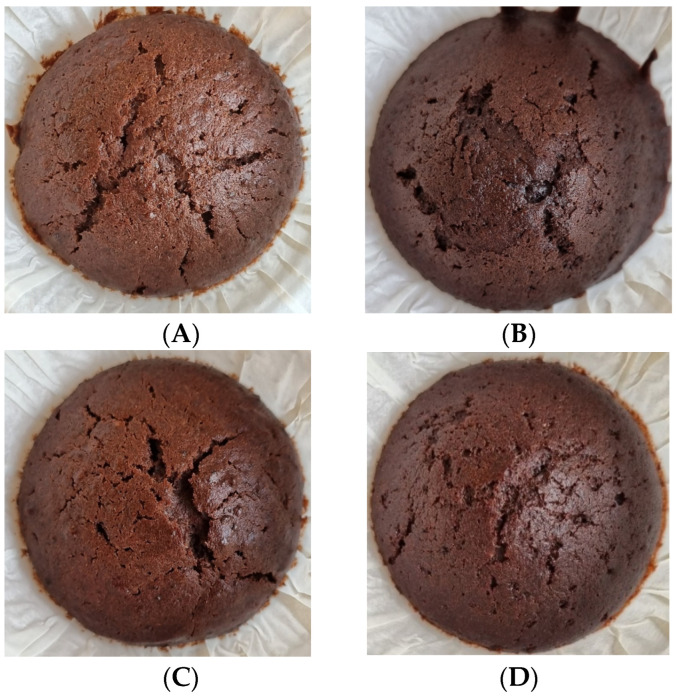
(**A**) Muffin control, (**B**) standard muffin with egg, (**C**) muffin with WAPC SFE treated, (**D**) muffin with WAPC not SFE treated.

**Table 1 foods-12-00845-t001:** Composition of APC, Welpro (WAPC), soy meal flour (SMF) and soy protein isolate (SPI). Data on SMF obtained from two sources (^a^ = [5], ^b^ = [6]). DM = dry matter.

Component [Reference]	APC [1]	Welpro (WAPC) [7]	SMF [5,6]	SPI [8]
Protein (% DM)	45–60	88.7	40.7 ^b^–49 ^a^	90.5–92.2
Protein digestibility (%)	63.6 [9]	91.7 [10]	86 ^a^	94.4–97.8
Fat (% DM)	9–11	<0.5	22.2 ^b^	0.2–1.2
Fibre (% DM)	11–15	<0.5	10.4 ^b^	-

**Table 2 foods-12-00845-t002:** Abbreviations of the four different types of white alfalfa protein concentrate produced and investigated in this study.

Treated with Supercritical CO_2_	Type of Press	
	Single Screw	Twin Screw
Not treated	WS	WT
Treated	WSS	WTS

**Table 3 foods-12-00845-t003:** Protein concentration, dry matter (DM), fat and L-a-b colour of the WAPC produced (from the twin screw press-based process), *n* = 2. Here, WT is WAPC without SFE treatment and WTS is WAPC treated with SFE.

Component	WT	WTS
Protein(% DM)	57.00 ± 0.44	64.56 ± 0.43
Dry matter (%)	93.83 ± 0.69	90.85 ± 1.26
Fat (% DM)	0.12 ± 0.16	0.00 ± 0.00
L-a-b colour	82.0, −6.9, 17.5	80.6, −6.9, 18.3

**Table 4 foods-12-00845-t004:** Composition of the raw alfalfa, the two types of WAPC (produced from the single screw press-based process), *n* = 2. Here, WS is not SFE treated; WAPC and WSS are SFE treated WAPC. NA = not analyzed or not applicable.

Component	Alfalfa	WS	WSS
Protein (% DM)	20.90 ± 2.42	57.73 ± 0.84	63.61 ± 0.07
Protein Digestibility (% DM)	NA	93.28 ± 0.89	92.65 ± 0.64
Soluble Protein (% of protein)	NA	14.98 ± 0.24	15.80 ± 0.42
Dry matter (%)	20.621	100 ± 0.00	99.8 ± 0.00
Ash (% DM)	8.61 ± 1.02	0.91 ± 0.14	Not determined
Fat (% DM)	1.22 ± 0.00	0.23 ± 0.25	0.00 ± 0.00
L-a-b colour	NA	63.6, 0.7, 26.5	61.3, 3.0, 22.0

**Table 5 foods-12-00845-t005:** Comparison of foaming properties of WAPC (WS) and WAPC SFE (WSS) compared to semi-skimmed milk, *n* = 1.

Sample	Force (N)	Foam Height (mm)
**Milk**	0.061	20.49
**WS**	0.256	38.766
**WSS**	0.251	34.117

**Table 6 foods-12-00845-t006:** Height and hardness of meringues produced with egg white, WT and WTS, *n* = 3; *p* > 0.05.

Protein Source	Height (mm)	Hardness (N)
**Egg white**	15.6 ± 0.6	27.3 ± 0.2
**WT**	13.0 ± 0.6	8.1 ± 0.3
**WTS**	12.8 ± 0.6	26.5 ± 3.56

**Table 7 foods-12-00845-t007:** Number of focus group participants detecting the taste of grass in the meringues produced with different protein ingredients.

Type of Meringue	No Grass	Mild Grass	Strong Grass
**Egg white**	19	0	0
**WT**	0	0	19
**WTS**	0	18	1

**Table 8 foods-12-00845-t008:** Texture and height of chocolate muffins baked with WS, WSS or egg, *n* = 3. Control muffin has no egg.

Type of Muffin	Hardness (N)	Chewiness (N)	Springiness (%)	Height (mm)
**Muffin control**	229.18 ± 34.48	41.83 ± 9.35	36.09 ± 2.91	23.76 ± 0.92
**Std. Muffin w. Egg**	135.42 ± 32.16	38.04 ± 12.47	58.57 ± 3.46	29.43 ± 2.43
**Muffin w. WS**	217.56 ± 65.34	34.05 ± 15.54	36.45 ± 5.13	28.13 ± 3.16
**Muffin w. WSS**	239.89 ± 47.53	36.03 ± 10.55	35.80 ± 2.49	27.02 ± 3.14

**Table 9 foods-12-00845-t009:** Intensity of grass taste in the different muffin types reported by the focus group.

Type of Muffin/Taste	No Grass	Mild Grass	Strong Grass
**Muffin control**	11	0	0
**Std. Muffin w. Egg**	11	0	0
**Muffin w. WS**	0	0	11
**Muffin w. WSS**	0	11	0

## Data Availability

The data is included in the article.

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
