# Peer review of "Treatment with Supercritical CO2 Reduces Off-Flavour of White Alfalfa Protein Concentrate"

_foods, 2023, doi:10.3390/foods12040845_

Round 1
Reviewer 1 Report
Dear Authors,
Apart from providing useful information about potential substitute for milk and egg protein you succeeded to significantly reduce the number of process steps. There are only two suggestions, one regards to better explanation about which drying technique (vacuum drying, liophilization, etc.) is applied on WAPC, while other regards to extraction time of 70 min (15 min static mode and 60 min dynamic mode)?
Kind regards
Author Response
The reviewer states: Apart from providing useful information about potential substitute for milk and egg protein you succeeded to significantly reduce the number of process steps. There are only two suggestions, one regards to better explanation about which drying technique (vacuum drying, liophilization, etc.) is applied on WAPC, while other regards to extraction time of 70 min (15 min static mode and 60 min dynamic mode)?
Thank you for your comments and relevant suggestion.
- The drying was only done by freeze drying, as described in line 123-126
“The white protein pellet was then collected and freeze-dried (Thermo Scientific Heto Drywinner DW8, Waltham, USA), starting at -20 °C with a 1 °C/h increase until the temperature reached 20 °C where it was kept, until no decrease in weight was observed.”
- Thank you for noting this. It was 75 min total. This is corrected
Line 144 now reads: Extraction was done for 75 min using either 180, 220 or ….

Reviewer 2 Report
In this work, a novel method of isolating protein from alfalfa is tested which includes a SFE stage to remove an unwanted grass taste. The method is much simpler than an established Pro-Xan II method, 9 stages vs. 19. Several white alfalfa protein concentrates were manufactured using the proposed procedure, with and without SFE treatment. Numerous tests revealed that the method gives a comparable, if slightly lower, productivity for white protein. Two types of baked goods were produced to test the applicability of alfalfa protein as a substitute for egg white as a foaming agent and emulsifier. Using muffins as a testing object, it was shown that the protein gave poor results as a substitute for egg as an emulsifier in muffin baking, supposedly, due to the absence of saponins. In case of meringues, alfalfa protein served as a good substitute for egg white as a foam-creating agent. It was shown that SFE treatment successfully diminishes grass taste from strong to mild, making baked good acceptable for consuming.
The work is very thorough, the aim of the study is relevant and within the scope of the journal, the results and novel and interesting. The manuscript is informative and an enjoyable reading. It should be accepted for publication with minor corrections enlisted below:
Remarks:
1) The main purpose of using SFE was to get rid of the unwanted grass taste, for which, according to authors, saponins such as zahnic acid are responsible. Results presented in Tables 7 and 9 show that this goal was achieved only partially: the grass taste was diminished, but not fully removed. In this regard, why was pressure in SFE held at only 220 bar? MV-10 system allows working at least at 350 bar. Higher extraction pressure would give higher solvating strength of supercritical CO2 and thus better extraction of saponins. In line 157, authors state that this pressure value was chosen as a compromise between extraction of chlorophyll and saponins. Why was it important to leave chlorophyll in the product? Plus, as can be seen from Fig. A2 in Appendix B, level of chlorophyll extraction was the same at 220 and 260 bar, so no risk of chlorophyll overextraction would seem to appear from rising pressure.
2) Section 2.2, SFE conditions. Was WAPC crushed or grinded after drying prior to SFE? If so, what was mean particle size? Please, specify.
3) Lines 371-373: single screw press is mentioned twice in this sentence. Apparently, it’s a mistyping, please correct.
4) Lines 465-468 and Table 6: a very interesting observation on the hardness of meringues produced with SFE-treated protein being three times higher than that of a non-treated one. Do authors have any tentative explanation of this observation?
5) Lines 500-503 and Table 8: authors suppose that saponins absence is responsible for low springiness of muffin made with alfalfa protein. But, according to Table 8, there is virtually no difference in mechanical properties between WS and WSS samples, whereas SFE presumably removes saponins from the protein, so there should be some difference. How do authors explain it? Were there actual measurements of saponin content in SFE-treated and untreated protein sample?
6) There are some minor mistypings in the text:
- the word “supercritical” should be written in one word throughout the text;
- line 23, gasses
- line 189, “50 mg of WT was mixed”
- line 433: probably, Table 5 was meant instead of Table 8.
- line 497: probably, Table 8 was meant instead of Table 9.
- Fig. A2 in Appendix B. Since authors made it a different appendix, perhaps, it is more prudent to name the figure B1 instead of A2.
Author Response
- The main purpose of using SFE was to get rid of the unwanted grass taste, for which, according to authors, saponins such as zahnic acid are responsible. Results presented in Tables 7 and 9 show that this goal was achieved only partially: the grass taste was diminished, but not fully removed. In this regard, why was pressure in SFE held at only 220 bar? MV-10 system allows working at least at 350 bar. Higher extraction pressure would give higher solvating strength of supercritical CO2and thus better extraction of saponins. In line 157, authors state that this pressure value was chosen as a compromise between extraction of chlorophyll and saponins. Why was it important to leave chlorophyll in the product? Plus, as can be seen from Fig. A2 in Appendix B, level of chlorophyll extraction was the same at 220 and 260 bar, so no risk of chlorophyll overextraction would seem to appear from rising pressure.
- Thank you for noting this. You are correct that the MV-10 is quite versatile and can, according to the manufacturer, work up to 350 bar. My own experience is that working above 330 bar makes the pressure less stable. So when working with WAPC we chose to work below 330 bar. However as you noted we did not investigate WAPC treated with SFE at higher pressures than 220 bar, with respect to food formulation. The main scope of the study was make a simple and cost efficient method so since we observed no difference from 220 to 260 we did not consider higher pressures, and instead, we chose 220 bar to lower the potential cost of the extraction. This is not clear in the text and we have added the following:
- Lines 373-377: “No difference was observed with respect to chlorophyll extraction when the pressure was increased from 220 to 260 bar. Therefore, 220 bar was chosen to minimize potential cost for a full-scale production. Future studies should include WAPC extracted at higher pressures to investigate if this could lower the taste of grass for the consumer, while maintaining the functionality of the WAPC”
- In line 157 we try to obtain a compromise between grass aroma, based on sniffing and chlorophyll, not saponins.
“ The level of grass aroma was observed by sniffing the protein powder after extraction. The results showed (See Appendix B) that a combination of 220 bar without the use of co-solvents at 45 °C led to the best compromise between removal of chlorophyll and grass aroma. These conditions were used in all future work.”
- Section 2.2, SFE conditions. Was WAPC crushed or grinded after drying prior to SFE? If so, what was mean particle size? Please, specify.
- Thank you for noting this. The WAPC was crushed by hand in a mortar and pestle and a particle size <1.25 mm was obtained by sieving. We have added a table as Appendix C with size distribution and the following text:
L126-128: “To ensure a uniform distribution during SFE treatment, manual crushing to a particle size below 1.25 mm (particle size distribution in Appendix C) was done with mortar and pestle.”
- Lines 371-373: single screw press is mentioned twice in this sentence. Apparently, it’s a mistyping, please correct.
-It was a comparison of the twin and single screw, this is corrected. Thank you for noting this.
- Lines 465-468 and Table 6: a very interesting observation on the hardness of meringues produced with SFE-treated protein being three times higher than that of a non-treated one. Do authors have any tentative explanation of this observation?
- Good point. We have added the following text:
L471-473: “The hardness of the WT meringue was lower than the other two batches, and this is speculated to be due the higher fat content, as described in Table 3. However, further investigation is needed to confirm this.”
- Lines 500-503 and Table 8: authors suppose that saponins absence is responsible for low springiness of muffin made with alfalfa protein. But, according to Table 8, there is virtually no difference in mechanical properties between WS and WSS samples, whereas SFE presumably removes saponins from the protein, so there should be some difference. How do authors explain it? Were there actual measurements of saponin content in SFE-treated and untreated protein sample?
- Unfortunately we did not measure the saponin content in any of our tests, this is only based on theory. We have added the following text:
L509-510: “However, the saponin content was not measured and future studies should be done to verify this.”
6) There are some minor mistypings in the text:
- the word “supercritical” should be written in one word throughout the text;
- line 23, gasses
- line 189, “50 mg of WT was mixed”
- line 433: probably, Table 5 was meant instead of Table 8.
- line 497: probably, Table 8 was meant instead of Table 9.
- Fig. A2 in Appendix B. Since authors made it a different appendix, perhaps, it is more prudent to name the figure B1 instead of A2.
-Thanks for finding these errors. All of them are corrected beside, “50 mg of WT was mixed” and the nomenclature in the Appendix since it is the standard suggested by MDPI.
